# Unveiling a magnetized jet from a low-mass protostar

Chin-Fei Lee [1,2], Hsiang-Chih Hwang[2,3], Tao-Chung Ching[4], Naomi Hirano[1], Shih-Ping Lai[4], Ramprasad Rao[1] & Paul.T.P Ho[1,5]

Protostellar jets are one of the most intriguing signposts in star formation. Recent detection of a jet rotation indicates that they can carry away angular momenta from the innermost edges of the disks, allowing the disks to feed the central protostars. In current jet-launching models, magnetic fields are required to launch and collimate the jets, however, observationally, it is still uncertain if magnetic fields are really present in the jets. Here we report a clear detection of SiO line polarization in the HH 211 protostellar jet. Since this line polarization has been attributed to the Goldreich-Kylafis effect in the presence of magnetic field, our observations show convincingly the presence of magnetic field in a jet from a low-mass protostar. The implied magnetic field could be mainly toroidal, as suggested in current jet-launching models, in order to collimate the jet at large distances.

[1] Academia Sinica Institute of Astronomy and Astrophysics, P.O. Box 23-141, Taipei 106, Taiwan. [2] Graduate Institute of Astronomy and Astrophysics, National Taiwan University, No. 1, Sec. 4, Roosevelt Road, Taipei 10617, Taiwan. [3] Department of Physics and Astronomy, Johns Hopkins University, Baltimore, MD 21218, USA. [4] Institute of Astronomy and Department of Physics, National Tsing Hua University, Hsinchu 30013, Taiwan. [5] East Asian Observatory, 660N. A'ohoku Place, University Park, Hilo, HI 96720, USA. Correspondence and requests for materials should be addressed to C.-F.L. (email: cflee@asiaa.sinica.edu.tw)

Protostellar jets are believed to be the key to removing angular momenta from the innermost edges of the accretion disks, allowing the disk material to fall onto the protostars during the star formation. Recent detection of a rotation in a protostellar jet at high angular and velocity resolutions strongly supports this role of the jets[1]. In current jet-launching models, the jets are launched from the accretion disks by magnetic fields through magneto-centrifugal force[2,3]. In these models, poloidal fields are required to launch the jets and toroidal fields are required to collimate the jets. Therefore, the jets are expected to be magnetized. However, there are still no observations showing convincingly the presence and morphology of magnetic fields in the jets.

In a few protostellar jets from massive protostars, for example, HH 80-81[4] and W3(OH)[5], their centimeter radio emission has been interpreted as synchrotron radiation because of its negative spectral index, as seen in the jets from active galactic nuclei (AGNs). Since synchrotron radiation is emitted by relativistic electrons in the presence of magnetic fields, this interpretation of radio emission suggests that the jets are magnetized. Detection of polarized radio emission can further support this interpretation, because synchrotron radiation is expected to be highly polarized. In addition, the observed polarization pattern can be used to derive the magnetic field morphology. So far, polarized radio emission was only detected in HH 80-81[6]. It was detected towards the jet-like structures at a large ($\gtrsim$30,000 a.u.) distance from the central protostar at a spatial resolution of ~20,000 a.u., with the polarization orientations roughly perpendicular to the jet axis. The implied magnetic field there could be helical, similar to what is expected in the AGN jets. A recent study with observations at ~8 times higher resolution suggested that the synchrotron emission there arises from an extended component and from the termination points of the jet, where strong shocks against the ambient medium can produce efficient particle acceleration[7].

The jets from low-mass protostars are less energetic than those from massive protostars, and thus only very few of them, for example, L778 VLA5[8] and DG Tau[9], show non-thermal emission that could be synchrotron radiation. Fortunately, when they are young, they have high content of molecular gas because of high mass-loss rate[10–13]. Therefore, a plausible way to map the magnetic fields in these jets is to map the linear polarization raised by the molecules (e.g., SiO and CO) themselves—the so-called Goldreich-Kylafis (GK) effect[14,15], as was previously done for molecular outflows[16]. In the presence of a magnetic field, a molecular rotational level splits into magnetic sublevels, producing a line polarization with its orientation either parallel or perpendicular to the magnetic field. In general, maximum polarization occurs when the line of sight is perpendicular to the velocity flow and the magnetic field direction[17]. Therefore, jets lying close to the plane of sky have the highest chance to produce observable polarization from the GK effect.

HH 211 is a nearby jet in the Perseus molecular cloud at a distance recently updated to be ~321 ± 10 pc[18]. It lies within ~9° of the plane of the sky[19,20], providing the best chance to map for the magnetic field. It is powered by a very young Class 0 protostar, which has a mass of $\lesssim 0.05 M_\odot$ [21,22] and is surrounded by a small rotating disk[22], and thus has a high content of molecular gas. It is best seen in SiO, appearing as a fast-moving highly collimated structure propagating down the jet axis inside the cavity of a slow-moving less-collimated outflow seen in $H_2$[23] (see Fig. 1) and CO[19,24]. It consists of the material coming out from the disk[19,23] and thus is intrinsically different from the CO outflow that consists mostly of ambient material. The SiO emission of the jet is brightest at $J = 8–7$ transition. At this transition, the optical depth is close to 1[19] and the collision rate is lower than the radiative transition rate for a typical jet density of $10^6–10^7$ cm$^{-3}$,

both optimal for polarization from the GK effect[14,15]. Here we report our detection of SiO line polarization in the inner part of the jet with the Atacama Large Millimeter/submillimeter Array (ALMA). This is the first convincing polarization detection towards the jet of a low-mass protostar, and at much closer distances from the protostar than in previous studies.

## Results

**SiO line polarization in the HH 211 jet.** Figure 2a shows the mean intensity map of the jet in the inner part in SiO $J = 8–7$ at $0''.21 \times 0''.14$ (67 a.u. × 45 a.u.) resolution. It shows the intensity averaged over a velocity range of ~30 km s$^{-1}$, in which the jet emission is detected, for both redshifted and blueshifted jet components. In our observations, the field of view (see Methods) is a circular region within ~8″ (i.e., 2600 a.u.) of the central source, covering the jet emission out to knots BK3 and RK3. The jet is knotty and highly collimated, as seen before. The linear structures within ~3″ of the central source are knots BK1 and RK1, and they are now each resolved into a few subknots, for example, A and B in knot BK1, as shown in Fig. 2b. The jet is marginally resolved in the transverse direction. The cylindrical radius of the jet, if assumed to be a half of the Gaussian deconvolved width of the subknots, is estimated to be $\lesssim$25 a.u. (i.e., $0''.08$, similar to that found before at a similar resolution[19]).

In the blueshifted jet component (blue contours), polarized SiO emission is clearly detected with $4\sigma$ to $7\sigma$ at the two brightest subknots, A and B, within ~1″.5 (~480 a.u.) of the central source and $3\sigma$ to $4\sigma$ at some other spots further out (see Fig. 2b). The polarization degree is about 1 to 2%, with a mean of ~1.5%, as shown by the length of the line segments in Fig. 2c. In addition, subknot A has a higher polarization degree than subknot B, probably because its optical depth is closer to 1[15]. Interestingly, all the polarization orientations are almost parallel to the jet axis. We can also map the polarization of the two subknots with enough signal-to-noise ratio in two wide-velocity channels with a width of ~14 km s$^{-1}$, as shown in Fig. 3. As can be seen, the polarization orientations and polarization degrees of the subknots in these two velocity channels are roughly the same and similar to those seen in Fig. 2c, which is obtained by averaging over the whole velocity range. This suggests that the polarization orientations and degrees are roughly the same at different velocities. Further observations with enough sensitivity in narrower velocity channels are needed to confirm this.

In the redshifted jet component (red contours in Fig. 2), the SiO emission is weaker than that in the blueshifted jet component. Assuming the same polarization degree of ~1.3% as in subknot B in the blueshifted jet component, the polarized intensity at the brightest subknots (peaks) is expected to be ~2 mJy per beam, which is only ~$3\sigma$. Also, since the subknots have a size much smaller than the maximum recoverable scale (~1″.1, see Methods) in our observations, the polarized emission, if it exists towards the subknots, will not be resolved out. Thus, deeper observations are needed to detect SiO line polarization in the redshifted jet component.

Our observations provide the first reliable detections of SiO line polarization in the protostellar jet, with high enough signal-to-noise ratios. Previously at four times lower resolution with the Submillimeter Array, we had two tentative detections of SiO line polarization in the outer part of the jet, one at knot RK2 and the other in between knot BK2 and BK3[25]. The one at knot RK2 has a polarization orientation parallel to the jet axis, as seen here in the inner part of the blueshifted jet component, but with a much higher polarization degree of ~10%. The one in between knots BK2 and BK3, where the SiO emission is faint, has a polarization degree of $\gtrsim$20% and an orientation inclined by ~50° to the jet

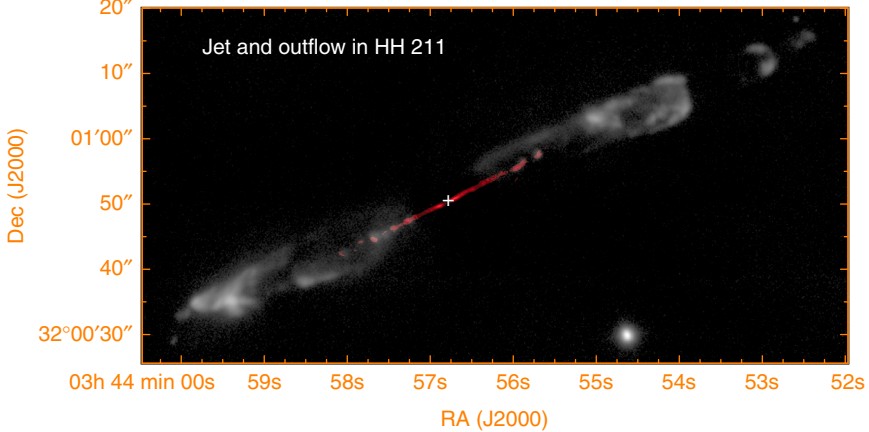

**Fig. 1** Jet and outflow in HH 211. Gray image shows the outflow and the outer part of the jet in $H_2$ adopted from Hirano et al.[23]. Red image shows the inner part of the jet in SiO adopted from Lee et al[19]. The cross marks the position of the central driving source at $\alpha_{(2000)} = 03^h43^m56.^s8053$, $\delta_{(2000)} = 32°00'50.''192$[22]

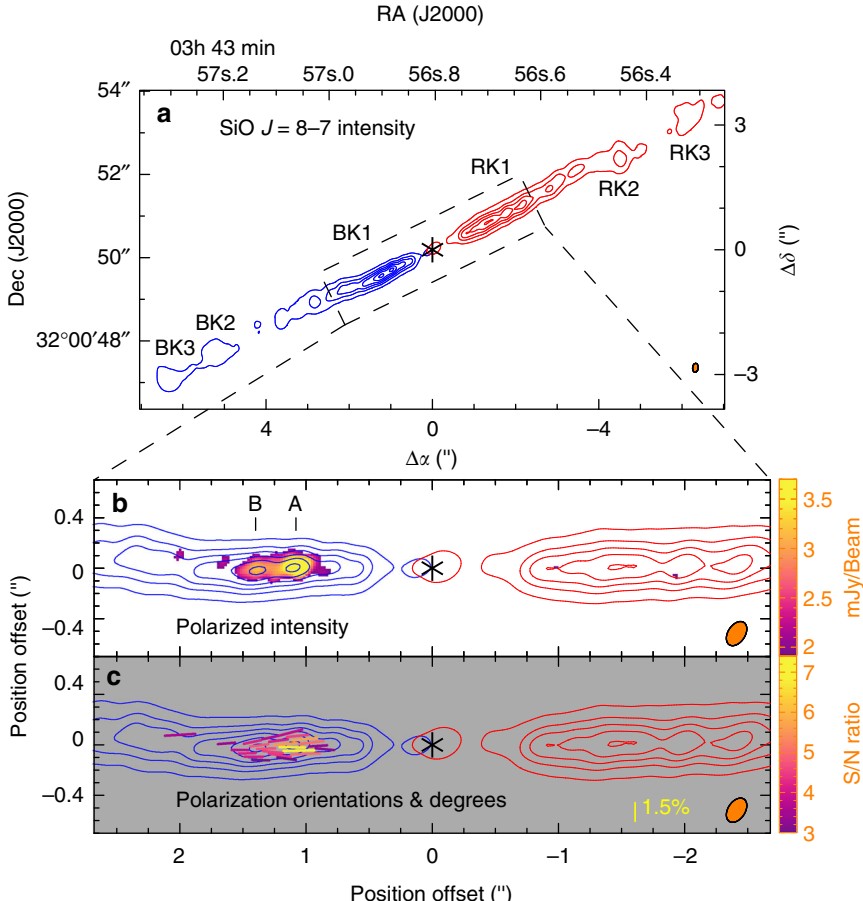

**Fig. 2 a** Polarization results towards the inner part of the HH 211 jet in SiO $J = 8$–$7$. The resolution is $0''.21 \times 0''.14$, as shown with the elliptical beams in the bottom right corners. Asterisks mark the source position. In **a**, knots BK1, BK2, BK3, RK1, RK2, and RK3 are the SiO knots identified before[11, 19]. Subknots A and B, where polarized emission is detected, are the two subknots in knot BK1 resolved in this work. In **b**, **c**, the image of the jet is rotated by 26.6° clockwise to be aligned with the x-axis. The blueshifted component (blue contours) is obtained by averaging the emission from −22 to 8 km s$^{-1}$, and the redshifted component (red contours) from 11 to 42 km s$^{-1}$. The contours start from $30\sigma$ with a step of $60\sigma$, where $\sigma \sim 0.64$ mJy per beam, which is the noise level of polarized emission. **b** The color image shows the polarized intensity $>3\sigma$ detection. **c** Line segments show the polarization orientations (E vectors), with their length indicating the polarization degree. They are color coded according to their S/N ratio in polarized intensity

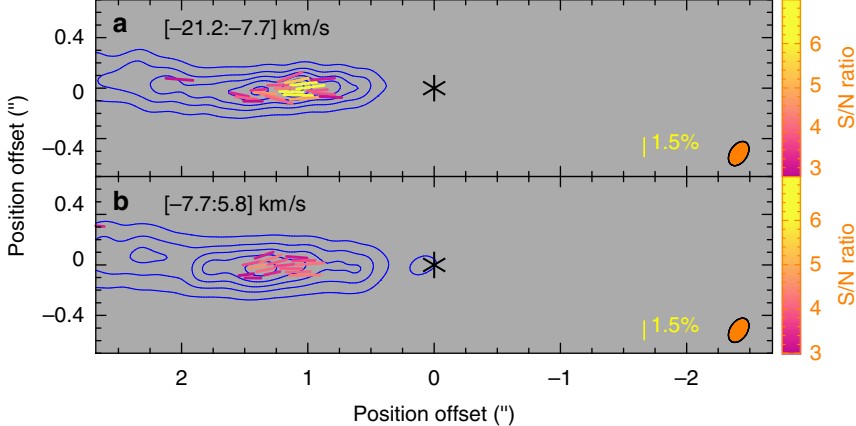

**Fig. 3** SiO line polarization results in the blueshifted component of the HH 211 jet in two wide-velocity channels. The velocity ranges are indicated in the upper left corners. The asterisk marks the position of the central driving source. Contours show the intensity of the SiO emission. They start from $30\sigma$ with a step of $60\sigma$, where $\sigma \sim 0.85$ mJy per beam, which is the noise level of polarized emission. Line segments show the polarization orientations (*E* vectors), with their length indicating the polarization degree. They are color coded according to their S/N ratio in polarized intensity

axis. Since those detections were only at ~$3\sigma$ level, further observations are needed to confirm them. Unfortunately, we cannot confirm those detections because we do not detect any polarized emission towards those knots. Notice that those knots are located more than 3″ away from the central source and thus outside the region where the polarization can be mapped properly in our observations (see Methods).

**Magnetic field morphology in the HH 211 jet**. SiO line polarizations have been detected in the circumstellar envelopes of evolved stars, with a maximum polarization degree of ~4%[26], higher than that seen here in the jet, probably because of different physical environments and velocity structures. They were attributed to the GK effect and thus used to map the magnetic field morphology in the circumstellar envelopes of evolved stars. Similarly, SiO line polarization detected here can be used to infer the magnetic field morphology in the jet. Here the polarization orientations are all almost aligned with the jet axis and thus the velocity flow axis. In this case, according to the GK effect, the magnetic fields could be either parallel or perpendicular to the polarization orientations[27], and thus could be either poloidal or toroidal.

One way to resolve this ambiguity in field morphology is to map one additional SiO line, because the polarization direction and relative polarization degree in two SiO lines could be used to judge the magnetic field direction, as done before with two CO lines for the CO outflows in other sources[16,28]. Another way to resolve the ambiguity is to compare the polarized SiO emission with polarized dust continuum emission in the same location, as done before to resolve the field ambiguity in the CO outflows in other sources[16,29,30]. Unfortunately, in our ALMA observations of HH 211, the dust emission at 350 GHz, which traces the envelope and disk, is only detected within ~0″.4 of the central source, without any overlap with the polarized SiO emission in the jet, as shown in Fig. 4. Previous observation of HH 211 at ~4″ resolution[31] showed extensions of dust continuum emission at 230 GHz along the outflow axis. Those extensions appear because of the accidental inclusion of CO emission in their map (C. Hull, private communication) and are thus probably from the CO outflow around the SiO jet. Further observations at higher sensitivity in continuum are needed to check if the dust emission can trace the jet and thus can be used to resolve the field ambiguity in the jet.

**Discussion**
Protostellar jets are generally thought to be launched from accretion disks[32]. In particular, two competing models, the X-wind model[2] and disk-wind model[3], have been constructed to launch the jets from the disks through the magneto-centrifugal force. The jets are launched from the disks at a radius of ~0.05 a. u. in the X-wind model[2], while from ~0.1 a.u. to a few a.u. in the disk-wind model[3]. In these two models, the magnetic fields in the jets at a distance much farther than the launching radius are expected to be mainly toroidal in order to collimate the jets[2,3]. Therefore, the magnetic fields at subknots A and B, which are at a distance of ~350 and 460 a.u. and thus much farther than the launching radius, should be mainly toroidal. If this is the case, the implied magnetic field orientations here would be perpendicular to the observed polarization orientations, as shown in Fig. 5. Since the observed orientations of the polarization give the projections of the magnetic fields on the plane of the sky, the magnetic fields here could also be helical, as suggested in the HH 80-81 jet from the massive protostar and in the AGN jets, but tightly wound. Further observations are needed to confirm this.

The magnetic field strength can be estimated in the two sub-knots, A and B, in the blueshifted jet component, where sufficient number of polarization vectors are detected. According to Chandrasekhar et al.[33] and Ostriker et al.[34], the field strength in the plane of the sky can be estimated with the following equation:

$$B \sim 0.5\sqrt{4\pi\rho}\frac{\Delta v_{\mathrm{los}}}{\Delta\phi}, \qquad (1)$$

where $\rho$ is the mass density, $\Delta v_{\mathrm{los}}$ is the velocity dispersion along the line of sight, and $\Delta\phi$ is the dispersion of the polarization angle. According to Lee et al.[35], the volume density in the jet $n_{\mathrm{H_2}} \gtrsim 2.0 \times 10^6$ cm$^{-3}$ (updated for the new jet radius because of new distance), and thus $\rho = 1.4 n_{\mathrm{H_2}} m_{\mathrm{H_2}} \gtrsim 9.3 \times 10^{-18}$ g cm$^{-3}$. Also from Figure 8a in Lee et al.[19], we have $\Delta v_{\mathrm{los}} \sim 15$ km s$^{-1}$, which is the full width at half-maximum of the SiO velocity distribution towards knot BK1. This velocity width can be produced by a turbulent motion due to localized shocks in the knot. From our observations, we have $\Delta\phi \sim 30° \pm 20°$, and thus $B \gtrsim 15^{+30}_{-6}$ mG.

Is this magnetic field strength reasonable? Answering this question requires us to know which jet-launching model is more appropriate for HH 211. Previously, a hint of jet rotation was reported in HH 211[19]. However, our observations here with

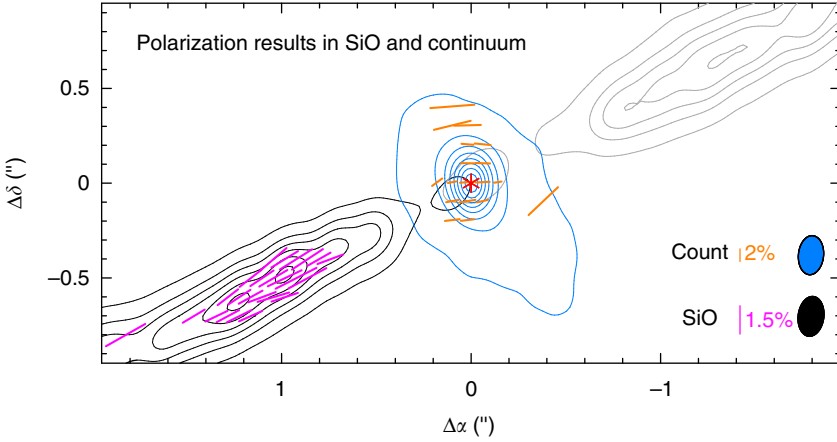

**Fig. 4** A comparison of polarized SiO emission with polarized dust continuum emission in HH 211. The asterisk marks the position of the central driving source. Black and gray contours show the blueshifted and redshifted jet components, respectively, with their contour levels the same as in Fig. 2. Magenta line segments show the polarization orientations of the polarized SiO emission. Blue contours and orange line segments show the intensity and the orientations of the polarized emission in continuum at 350 GHz, obtained alongside in our ALMA observations (Lee et al., in preparation)

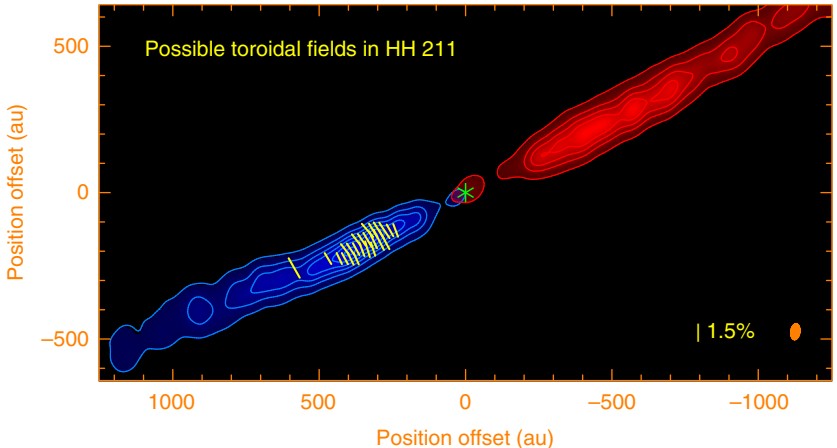

**Fig. 5** Possible toroidal fields in the HH 211 jet. Yellow line segments indicate the possible toroidal fields obtained by rotating the polarization orientations by 90°. The asterisk marks the position of the central driving source. Blue image with the blue contours shows the blueshifted jet component. Red image with the red contours shows the redshifted jet component. The contour levels are the same as in Fig. 2

| Table 1 Correlator setup | | | | | |
| --- | --- | --- | --- | --- | --- |
| **Spectral window** | **Line or continuum** | **Number of channels** | **Central frequency (GHz)** | **Bandwidth (MHz)** | **Channel width (kHz)** |
| 0 | SO $N_J = 8_9 - 7_8$ | 480 | 346.533 | 117.187 | 244.141 |
| 1 | CO $J = 3 - 2$ | 480 | 345.800 | 117.187 | 244.141 |
| 2 | SiO $J = 8 - 7$ | 960 | 347.335 | 468.750 | 488.281 |
| 3 | HCO$^+$ $J = 4 - 3$ | 960 | 356.739 | 234.375 | 244.141 |
| 4 | Continuum | 960 | 358.005 | 1875.000 | 1953.125 |

similar resolution but much higher sensitivity found no clear rotation. Therefore, with the velocity (~0.42 km s$^{-1}$ per channel) and spatial (~64 a.u.) resolutions in our observations, we can assume an upper limit of ~27 a.u. km s$^{-1}$ for the jet-specific angular momentum. Then, using a jet velocity[20,35] of ~150 km s$^{-1}$ and a protostellar mass of ~0.05 $M_\odot$[22], the jet-launching radius is estimated to be ≲0.06 a.u.[36] and thus more consistent with the X-wind model. In a typical X-wind model for a low-mass protostellar

jet-like HH 211, the magnetic field strength is expected to be ≳6 mG for a jet cylindrical radius of ≲25 a.u.[37]. Hence, the estimated field strength from our observations is about 2.5 times as high as the expected, probably reasonable considering a shock compression in the subknots and all the uncertainties in the measurements. The resulting magnetic pressure will be $B^2/8\pi \gtrsim 8.9 \times 10^{-6}$ dyne cm$^{-2}$. This is much higher than the thermal pressure in the subknots, which is $1.2 n_{\mathrm{H}_2} kT \gtrsim 1.7 \times 10^{-7}$ dyne

cm$^{-2}$ with a density $n_{H_2} \sim 2.0 \times 10^6$ cm$^{-3}$ and the highest possible $T \sim 500$ K[23]. As a result, the magnetic field, if toroidal, is strong enough to confine and thus collimate the jet material.

In conclusion, we have detected convincingly polarized SiO line emission in the protostellar jet, although deeper observations are needed to detect the polarization in the redshifted component of the jet and check for consistent morphology of the polarization. Our detection has opened up an opportunity to use the GK effect to map the magnetic field morphology in the jet in the early phase of star formation. Based on the current jet-launching models, the implied magnetic field should be mainly toroidal at large distance. In the future, we will follow-up with polarization observations in SiO $J = 5 - 4$ in order to confirm the field morphology.

## Methods

**Observations**. Polarization observations of the HH 211 protostellar jet were carried out with ALMA in Band 7 at ~350 GHz in Cycle 4, with 43 antennas in C40-7 configuration. The project number was 2016.1.00017.S. The phase center was 03$^h$43$^m$56.$^s$8040 $\delta_{(2000)} = 32°00'50.''270$, but the maps here are produced and presented with a center at the central source position at $\alpha_{(2000)} = 03^h43^m56.^s8053$, $\delta_{(2000)} = 32°00'50.''192$[22]. Two executions were carried out on 10 October 2016, with a total time of ~77 min on the HH 211 jet. The projected baselines are ~20–1500 m. One pointing, with a primary beam (field of view) of ~17″ covering a circular region within ~8″ of the central source, was used to map the inner part of the jet. The maximum recoverable scale is ~1″.1. Since the jet consists of a chain of knots and subknots, the detection of polarized emission towards the subknots, which have a size of ~0″.2, will not be affected. However, since the jet seems to have a smooth structure greater than the maximum recoverable scale, the polarized emission there, if exists, could be partially resolved out.

The correlator was set up to have five spectral windows, with one for SO $N_J = 8_9 - 7_8$ at 346.528481 GHz, one for CO $J = 3 - 2$ at 345.795991 GHz, one for SiO $J = 8 - 7$ at 347.330631 GHz, one for HCO$^+$ $J = 4 - 3$ at 356.734288 GHz, and one for the continuum at 358 GHz (see Table 1). In this paper, we only present the observational results in SiO, which traces uniquely the jet emanating from the central source. The velocity resolution is 0.42 km s$^{-1}$ per channel in SiO. The data were calibrated with the CASA package (versions 4.7), with Quasar J0238 + 1636 as a flux calibrator, Quasar J0237 + 2848 (~0.833 Jy) as a passband calibrator, Quasar J0336 + 3218 (~0.517 Jy) as a gain calibrator, and Quasar J0334 − 4008 (~ 0.415 Jy) as a polarization calibrator. A phase-only self-calibration of the data was performed to improve the map fidelity using the continuum intensity (Stokes I) map towards the central source. We used a robust factor of 2 for the visibility weighting to generate the SiO maps (including Stokes I, Q, and U parameters) at a resolution of ~0″.2. No primary beam correction is applied to the maps here because we only focus on the inner part of the jet where the polarized emission can be mapped properly (see below). In addition, no polarization is detected beyond 2″ of the central source. In order to detect the polarization at sufficient sensitivity, we made the mean intensity maps for the redshifted jet component and blueshifted jet component (see Fig. 2), by averaging the emission intensity over a velocity range of ~30 km s$^{-1}$, in which the jet emission is detected. The noise level is ~2.73 mJy per beam in the Stokes I map and ~0.64 mJy per beam in the Stokes Q and U maps for the SiO line. In order to check for any polarization dependence on velocity in the blueshifted jet component where the polarization is well detected, we also present polarization maps in two wide-velocity channels with a velocity width of ~14 km s$^{-1}$. The noise level is ~0.85 mJy per beam in the Stokes Q and U maps of those wide channels.

Polarization orientations are defined by the E vectors. Linear polarization degree (fraction) is defined as $P = \sqrt{Q^2 + U^2}/I$. According to ALMA Technical Handbook in Cycle 4, polarization imaging with accuracy better than 0.3% can be achieved within the inner 1/3 of the primary beam and this determines the largest acceptable angular size of sources which can be observed in full polarization. Therefore, in our observations with the phase center at the source position, we can only map the polarization of the jet reliably within ~3″ of the central source.

## Data availability

This article makes use of the following ALMA data: ADS/JAO.ALMA#2016.1.00017.S. The data that support the plots within this paper and other findings of this study are available from the corresponding author upon reasonable request.

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

## Acknowledgements

This paper makes use of the following ALMA data: ADS/JAO.ALMA#2016.1.00017.S. ALMA is a partnership of ESO (representing its member states), NSF (USA), and NINS (Japan), together with NRC (Canada), NSC and ASIAA (Taiwan), and KASI (Republic of Korea), in cooperation with the Republic of Chile. The Joint ALMA Observatory is operated by ESO, AUI/NRAO, and NAOJ. C.-F.L. acknowledges grants from the Ministry of Science and Technology of Taiwan (MoST 104-2119-M-001-015-MY3 and 107-2119-M-001-040-MY3) and the Academia Sinica (Career Development Award).

## Author contributions

C.-F.L. led the project, analysis, discussion, and drafted the manuscript. H.-C.H. produced a preliminary SiO polarization model from the GK effect to compare with our observations. C.-F.L., H.-C.H., T.-C.C., N.H., S.-P.L., R.R., and P.T.P.H. contribute to scientific discussion.

## Additional information

**Competing interests:** The authors declare no competing interests.

