## [Peer Review File · Nature Communications]

Reviewers' comments:

Reviewer #1 (Remarks to the Author):

I have reviewed the submission by Lee et al. entitled, "Unveiling a Magnetized Jet from a Low-Mass Protostar," and find it to be a concise report of the most convincing high-resolution detection of spectral-line polarization to date in a protostellar outflow/jet.

My main concern is the claim that this is the first detection of polarized emission from a protostellar jet, as the authors state in the text that the jet is actually smaller than the resolution of their ALMA data. This point should be clarified, as, if this is in fact a detection of polarized emission in ambient material, then this work could simply be considered a higher sensitivity, higher resolution version of other studies that have focused on polarized emission from lower-velocity protostellar outflows in the past.

In addition to a number of small comments, I also request an analysis of the continuum dust polarization map from the ALMA data alongside the SiO polarization. In the event that there is overlapping polarized dust & SiO emission, this will provide an important comparison with the SiO emission, and may even reveal what the actual magnetic field orientation is in the jet, which would help to resolve the ambiguity in the polarization from the GK effect.

I would like to see a revised version of the draft, with the new and updated text printed in bold.

COMMENTS

***** General *****

(1)

HH211 is a well known protostar that has been observed in full polarization at least once before (e.g., Hull et al. 2014). It would be useful if the authors could show a map of continuum dust polarization from their ALMA data to compare with the SiO polarization, in particular if there are any areas that overlap physically. It seems quite likely that there will be dust emission that overlaps with the SiO emission, based on the Stokes I contour map from the CARMA data, which shows extensions of dust emission (presumably from outflow-heated dust) along the orientation of the SiO jet.

Furthermore, the dust emission along the outflow axis is likely to be polarized, as even the limited-sensitivity CARMA data detected polarization along the redshifted outflow lobe. Unfortunately, the redshifted polarized dust emission cannot be compared with the SiO polarization, as the SiO polarization is undetected in the redshifted outflow. However, if dust polarization is detected on the blueshifted side -- and is consistent with a magnetic field oriented along the outflow direction, as is the case on the redshifted side -- then the interpretation of the GK polarization regarding poloidal vs. toroidal magnetic fields will have to be reconsidered.

This kind of dust vs. spectral-line polarization comparison has been performed in the past to boost confidence that -- at least in regions of overlapping spectral-line polarization and dust polarization -- the two methods are tracing the same magnetic field. For example: Girart et al. 1999 and Ching et al. 2016 in the outflow of NGC 1333-IRAS 4A; and Lai et al. 2003 in DR 21 (although the CO

emission observed there is not necessarily from an outflow).

(2)

From Figure 1, the jet/outflow is clearly resolved in your observations. However, you say in the text that the radius of the jet is <20 au. Are you thus saying that you're NOT resolving the jet, and rather are resolving ambient material? If so, then it seems that these observations may suffer from the same issue as the HH 80-81 observations mentioned on page 2 (i.e., that the polarization is from the ambient material, not from the jet itself).

I agree that this is by far the highest-resolution observation of spectral-line polarization in a jet/outflow ever detected -- but if it can't be argued that this emission is from the jet itself, then these observations are fundamentally no different from the previous observations of spectral-line polarization in, e.g., NGC 1333-IRAS 4A.

(3)

Is the signal-to-noise in the blueshifted polarization SiO emission high enough to investigate the polarized emission as a function of velocity? It would be good to know whether any information can be derived from changes in polarization angle & polarization fraction as a function of velocity, and whether the emission is smoothly distributed across the ~30 km/s range over which you integrated, or whether it is peaked strongly at one or more very specific velocities.

***** Abstract *****

"Since the line polarization has been attributed..."  are you saying this in a general sense, i.e., all spectral-line polarization at these frequencies can be attributed to the GK effect? Or are you claiming this only for this specific case?

***** Main text *****

Page 2

Change "excess angular momenta"  "angular momentum"

Page 3

Footnote 14 makes the text look like 1^{14} : please make that reference parenthetical. This applies elsewhere in the paper as well where footnotes are connected to numbers.

Page 4

Define BK1, BK3, RK1, RK3 and give a citation (if this isn't the first time these abbreviations have been used in this source; otherwise, reference Figure 1a).

Are there any reasons besides purely sensitivity why the redshifted component of the outflow would not have detectable polarization? Are there known physical differences between the two lobes of the outflow? It is particularly intriguing that the CARMA observations of the inferred magnetic field show much more polarized emission along the redshifted outflow lobe, where no SiO polarization is detected.

Page 6

Please describe where the velocity dispersion comes from: for the Davis-Chandrasekhar-Fermi method to make sense, this velocity dispersion should ideally be from turbulence, and thus it is important to know which tracer it is. If I am correct in assuming that the quoted ~ 15 km/s is the FWHM of the SiO velocity distribution in Figure 8 of Lee et al. 2009, then that is not necessarily a good tracer, since the SiO width is dominated by the outflow, not by turbulent motions that would be disturbing an ordered magnetic field.

Additionally: please state that $2.8e6$ /cc is the volume density of the jet (assuming that my interpretation of that number from Lee et al. 2010 is correct). Otherwise it might be confused with, for example, the volume density of the gas+dust in the dense core.

Page 7

You mention the very reasonable idea of following up with SiO(5-4). However, you say on page 3 that SiO(8-7) is the brightest transition in HH 211. How much fainter is SiO(5-4), and what is an approximate time estimate for how long it would take to get a robust ($>3\sigma$) detection of polarization in both the red- and blueshifted lobes in the HH 211 outflow in the fainter J=5-4 transition? Is it doable in what is now considered a "reasonable" amount of ALMA time?

***** Methods: Observations *****

¶2: the noise levels referred to are in the spectral-line data (not the dust continuum), correct? Best to say that specifically.

¶3: the note about ALMA's restrictions is correct. However, this statement makes me wonder: is all of the detected polarized emission within a radius of $\sim 2''$ from the source position? Or are you masking robustly detected polarization further out in the jet? If polarization IS detected far out in the jet (i.e., where you can compare with the 2014 SMA results), I believe it should still be shown, just with stated caveats regarding deviations in polarization position angle and polarization fraction further out in the ALMA primary beam.

Please describe how you performed self-calibration on the image, if you did in fact do so (and if you didn't, then you definitely should to increase image fidelity).

Please state whether the images in Figures 1 and 2 are primary-beam corrected.

***** Figures *****

FIGURE 1

-Mention in the figure caption what A, B, and the BKs/RKs are

-I recommend using a perceptually uniform sequential colorscale for both the polarized intensity colorscale and the line-segment colors

-Example: https://matplotlib.org/examples/color/colormaps_reference.html

-Please do not use light green or cyan on a white background: they are very hard to see

FIGURE 2

-Caption error: "Red image with the red contours shows the REDSHIFTED jet component."

-Please make sure that the line segments are plotted at the Nyquist frequency (max 2 per beam) along both the major and minor axes of the synthesized beam. It appears that the line segments are a bit too close together at the moment (though I may be wrong).

Reviewer #2 (Remarks to the Author):

This paper reports the detection and mapping of SiO line polarization towards the jet of a low-mass protostar. This result gives valuable information on the distribution of the magnetic field towards this jet, since this linear polarization is expected to be aligned either parallel or perpendicular to the projection of the magnetic field on the plane of the sky. This is one of the best cases of protostellar jets in which polarization has been mapped, and likely the best example in a low-mass protostar. It is also the case in which the polarization/magnetic field has been mapped in the jet at closest distances to the star (a few hundred au). These results would make the paper suitable for publication in Nature Communications. However, in its present form, the manuscript presents severe problems that should be fully addressed before consideration for publication.

Main issues:

1. The manuscript apparently reports observations of the HH 211 jet, located in the Perseus molecular cloud. The coordinates of the center of the HH 211 jet are approximately RA(J2000) = 03 43 57, DEC(J2000) = +32 00 50. However, all the images in the paper are given in offsets relative to the central source of the jet, and the absolute coordinates of this central source, which are given only in the caption of Fig 1, RA(J2000) = 05 43 51.4086, DEC(J2000) = -01 02 53.147, correspond to a completely different region of the sky. I concluded that the coordinates given in the paper actually correspond to the HH 212 jet. Thus, which one of the two objects, HH 211 or HH 212, has been observed? This is a serious error that compromises the credibility of the observational data presented in the paper.

Fortunately, the orientation of the two jets is very different. The HH 211 jet is oriented in a SE-NW direction while HH 212 is oriented in a SW-NE direction, suggesting that the images presented in the paper actually correspond to the right object. However, some authors prefer to rotate the HH 212 jet by 90 deg, in which case the two jets look very similar. The images presented in the paper do not indicate the direction of the Position Offsets, making more difficult the correct identification. Are the Offsets in the RA, DEC directions? Where is the North?

The authors must completely dissipate any doubt on the identification of the source of their observations before further consideration for publication of this paper. I suggest to use absolute

coordinates instead of offsets in at least one of the images presented. Further details on the observations and observational setup should be given in the Methods section (e.g. give the project number, position of the phase center, etc). The physical properties of the central protostar and its close environment are obtained from the literature, including papers with no absolute coordinates (in a number of cases with several authors in common with those of this paper); because the differences between the central sources are not as obvious as in the jets, it should be double-checked that the parameters adopted really correspond to the driving source of HH 211 and not to that of HH 212.

2. Related with the issue of the position of the source, the authors may want to know that a distance of 321 ± 10 pc has recently been proposed for IC 348, where the HH 211 jet is located (see Ortiz-Leon et al. 2018, arXiv:1808.03499v1). This estimate has been obtained from new VLBI data and seems to be in better agreement with Gaia DR2. Ortiz-Leon et al. suggest that the previous estimate by Hirota et al. (2011) (the value used by the authors, although not referenced), obtained from water masers, could be affected by variability and problems in the identification of the maser features. This issue must be discussed and adoption of the new distance should be considered.

3. The authors introduce their results in a way that may suggest that it is the first mapping of the polarization/magnetic field in a protostellar jet (3rd paragraph), and that there was no previous evidence of synchrotron radiation in a jet from a low-mass protostar (4th paragraph). This is not correct, as there were previous results providing convincing evidence of both things, and this should be properly explained.

Carrasco-Gonzalez et al. (2010) discuss in detail the strength, direction and spatial distribution of the magnetic field, and give quite convincing evidence that it is intrinsically associated with the HH80-81 jet. For example, Figure 3 in that paper shows that the magnetic field inferred from the polarization of the radio jet is very well aligned along the jet axis, while the direction of the magnetic field in the ambient cloud inferred from the dust emission presents a large dispersion. This information was derived from linearly polarized synchrotron emission in a way similar to the well-studied AGN jets. For non-relativistic jets, as is the case of the HH80-81 jet, such an approach has the advantage that the magnetic field projected on the plane of the sky is perpendicular to the polarization, with no ambiguity. Carrasco-Gonzalez et al. favor an helical geometry for the magnetic field, again similar to what is expected in AGN jets. Given the large size of the HH 80-81 jet, the angular resolution in that paper (corresponding to about 1500 au) is sufficient to map the distribution of the magnetic field with respect to the jet, but at distances $> 30,000$ au from the exciting source, as noted by the authors of the present manuscript. Therefore, the main novelty of the results presented here is that polarization is mapped for the first time towards the jet of a low mass protostar, and at much closer distances from the star than in previous studies.

The paper by Rodriguez-Kamenetzky et al. (2017) does not seem to contradict the previous paper by Carrasco-Gonzalez et al., as apparently is suggested in the manuscript (actually, both papers have many authors in common). The paper by Rodriguez-Kamenetzky et al. mainly deals with the origin of the relativistic electrons that produce the synchrotron emission rather than with the distribution of the magnetic field. Polarization is not detected in the second paper despite the increased sensitivity. The authors attribute this non-detection to the lower flux density per beam because of the higher angular resolution. Also, the large depolarization effects due to Faraday Rotation in the large bandwidth of the new observations likely have an important contribution. Therefore, the magnetic field distribution cannot be discussed in detail from the results of this second paper. Nevertheless, the higher angular resolution of Rodriguez-Kamenetzky et al. led these authors to suggest that the synchrotron emission in the jet arises from an extended component and from the termination points of the jet. In my opinion this does not exclude that the magnetic field mapped by Carrasco-Gonzalez et al. is intrinsically associated with the jet itself; otherwise, a re-orientation along the jet axis of the ambient cloud magnetic field would be

required. Sensitive high-angular resolution observations with a narrow bandwidth would set this issue.

I suggest the 3rd paragraph to be reworded to better reflect the available previous observations, in order to clarify the relevance of the new data.

In the 4th paragraph, it is not correct to say that "the jets from low-mass protostars do not show any signs of synchrotron radiation". Carrasco-Gonzalez et al. (2010) mention several examples of non-thermal emission in jets from low-mass objects suggestive of synchrotron. Also, the paper by Ainsworth et al. (2014), ApJ, 792, L18 presents tentative evidence for synchrotron emission associated with the jet of the young low-mass star DG Tau, and should be cited.

Minor Comments:

Paragraph 9: Only poloidal and toroidal magnetic fields are considered. What about an helical magnetic field? Note that what is obtained from the observed direction of the polarization is the projection of the magnetic field on the plane of the sky.

Paragraph 12: A "jet radius" of < 20 au is mentioned. It could be better to use "cylindrical radius" to avoid ambiguities with the launching radius or with the distance from the origin of the jet.

Methods: It is stated that the maximum recoverable scale is about 1.1 arcsec. This scale is larger than the width of the HH 211 jet but significantly smaller than the extension of both the blue and the red lobes along the direction of the jet axis ($> 3-5$ arcsec). How this observational constraint affects the results presented here? Is it feasible that the polarized emission of a possible extended (> 3 arcsec) structure elongated along the jet axis is partially resolved out by the interferometer? Could this effect "erase" part of the emission and affect the observed direction of the polarization vector? Could this effect be partially responsible for the observed lack of polarization in the redshifted lobe?

Caption of Fig 1: the jet is rotated...  the image of the jet is rotated..

Guillem Anglada

Dear Reviewers,

Thank you very much for your constructive comments. We have revised the paper accordingly and hope that our revision is adequate for publication. The changes are in bold-faces.

BTW, we have adopted the new distance of 321 pc and updated all the related quantities in our paper accordingly. E.g., the jet (cylindrical) radius is now $< \sim 25$ au, etc.

Best Regards,
Chin-Fei

=====
Reviewers' comments:

Reviewer #1 (Remarks to the Author):

I have reviewed the submission by Lee et al. entitled, "Unveiling a Magnetized Jet from a Low-Mass Protostar," and find it to be a concise report of the most convincing high-resolution detection of spectral-line polarization to date in a protostellar outflow/jet.

REVIEWER:

My main concern is the claim that this is the first detection of polarized emission from a protostellar jet, as the authors state in the text that the jet is actually smaller than the resolution of their ALMA data. This point should be clarified, as, if this is in fact a detection of polarized emission in ambient material, then this work could simply be considered a higher sensitivity, higher resolution version of other studies that have focused on polarized emission from lower-velocity protostellar outflows in the past.

ANSWER: With the new adopted distance, the jet now has a radius of $< \sim 25$ au or a diameter of 50 au. It is marginally resolved in our observations, which has a resolution of $0.21'' \times 0.15''$ or $67 \text{ au} \times 45 \text{ au}$. The SiO jet here is a fast-moving highly-collimated structure propagating down the jet axis inside the cavity of a slow-moving less-collimated outflow seen in H₂ (see new Figure 1) and CO. It consists of the material coming out from the disk and thus is intrinsically different from the outflow that consists mostly of ambient material. Thus, our polarization detection is the first convincing detection towards the jet from a low-mass protostar. We have made all these clear in the paper, adding a new Figure 1.

REVIEWER:

In addition to a number of small comments, I also request an analysis of the continuum dust polarization map from the ALMA data alongside the SiO polarization. In the event that there is overlapping polarized dust & SiO emission, this will provide an important comparison with the SiO emission, and may even reveal what the actual magnetic field orientation is in the jet, which would help to resolve the ambiguity in the polarization from the GK effect.

ANSWER: We have added the continuum dust polarization map obtained alongside the same ALMA data (see new Figure 4). As can be seen, there is no overlap between the polarized dust emission and the polarized SiO emission, and thus the continuum map can not be used to resolve the ambiguity in the SiO line polarization from the GK effect. A new paragraph is now added for this.

BTW, no analysis of the continuum map is added here because the continuum map does not help and adding any analysis would distract the readers from the jet polarization. Instead, the analysis of the continuum map will be presented in a future publication.

REVIEWER:

I would like to see a revised version of the draft, with the new and updated text printed in bold.

ANSWER: OK

COMMENTS

***** General *****

(1)

REVIEWER:

HH211 is a well known protostar that has been observed in full polarization at least once before (e.g., Hull et al. 2014). It would be useful if the authors could show a map of continuum dust polarization from their ALMA data

to compare with the SiO polarization, in particular if there are any areas that overlap physically. It seems quite likely that there will be dust emission that overlaps with the SiO emission, based on the Stokes I contour map from the CARMA data, which shows extensions of dust emission (presumably from outflow-heated dust) along the orientation of the SiO jet.

Furthermore, the dust emission along the outflow axis is likely to be polarized, as even the limited-sensitivity CARMA data detected polarization along the redshifted outflow lobe. Unfortunately, the redshifted polarized dust emission cannot be compared with the SiO polarization, as the SiO polarization is undetected in the redshifted outflow. However, if dust polarization is detected on the blueshifted side -- and is consistent with a magnetic field oriented along the outflow direction, as is the case on the redshifted side -- then the interpretation of the GK polarization regarding poloidal vs. toroidal magnetic fields will have to be reconsidered.

ANSWER: As mentioned above, we have added a new Figure 4 to show the

continuum map from our ALMA data to compare with the SiO polarization. However, we do not see any continuum emission extending along the SiO jet. Based on a private communication with the first author of the CARMA paper, the Stokes I contour map from the CARMA data actually included CO emission accidentally, and thus appeared to extend along the outflow axis overlapping with the SiO jet. Therefore, those extensions are probably from the CO outflow around the SiO jet. Further observations at higher sensitivity in continuum are needed to check if the dust emission can trace the jet and thus can be used to resolve the field ambiguity in the jet. A note is added.

REVIEWER:

This kind of dust vs. spectral-line polarization comparison has been performed in the past to boost confidence that -- at least in regions of overlapping spectral-line polarization and dust polarization -- the two methods are tracing the same magnetic field. For example: Girart et al. 1999 and Ching et al. 2016 in the outflow of NGC 1333-IRAS 4A; and Lai et al. 2003 in DR 21 (although the CO emission observed there is not necessarily from an outflow).

ANSWER: In all those sources, dust traces the envelope and CO traces the outflow (consisted mostly of ambient material) and envelope, and thus there are some overlaps between the dust and CO emission. Here dust emission and SiO emission trace different materials, with the dust emission tracing the envelope/disk and the SiO emission tracing the material emanating from the disk. Therefore, no overlap is seen, as mentioned above. A note has been added to emphasize this.

(2)

REVIEWER:

From Figure 1, the jet/outflow is clearly resolved in your observations. However, you say in the text that the radius of the jet is <20 au. Are you thus saying that you're NOT resolving the jet, and rather are resolving ambient material? If so, then it seems that these observations may suffer from the same issue as the HH 80-81 observations mentioned on page 2 (i.e., that the polarization is from the ambient material, not from the jet itself).

ANSWER: With the new distance the jet has a radius of $<\sim 25$ au or a diameter of $<\sim 50$ au. The resolution is $67\text{au} \times 48\text{au}$, and thus the jet is marginally resolved. As shown now in new Figure 1, the jet here is clearly different from the outflow seen in H2 and CO. It is the material coming out from the disk (see explanation above).

REVIEWER:

I agree that this is by far the highest-resolution observation of spectral-line polarization in a jet/outflow ever detected -- but if it can't be argued that this emission is from the jet itself, then these observations are fundamentally no different from the previous observations of

spectral-line polarization in, e.g., NGC 1333-IRAS 4A.

ANSWER: See above. We have clarified that the SiO jet is the material coming out from the disk, and thus intrinsically different from those outflows in previous studies. Those outflows consist mostly of (swept-up) ambient material.

(3)

REVIEWER:

Is the signal-to-noise in the blueshifted polarization SiO emission high enough to investigate the polarized emission as a function of velocity? It would be good to know whether any information can be derived from changes in polarization angle & polarization fraction as a function of velocity, and whether the emission is smoothly distributed across the ~30 km/s range over which you integrated, or whether it is peaked strongly at one or more very specific velocities.

ANSWER: The signal to noise ratio is good enough to map the SiO polarization towards the blueshifted jet component in two wide-velocity channels, as shown in new Figure 3. As can be seen, the polarization angles and polarization degrees in these two velocity channels are roughly the same and similar to those seen in Figure 2c, which is obtained by averaging over the whole velocity range. This suggests that the polarization orientations and degrees are roughly the same at different velocities. Further observations with enough sensitivity in smaller velocity channels are needed to confirm this.

***** Abstract *****

REVIEWER:

"Since the line polarization has been attributed..."  are you saying this in a general sense, i.e., all spectral-line polarization at these frequencies can be attributed to the GK effect? Or are you claiming this only for this specific case?

ANSWER: The sentence has been revised from

"Since the line polarization" ==> "Since this line polarization"

***** Main text *****

Page 2

REVIEWER:

Change "excess angular momenta"  "angular momentum"

ANSWER: "excess" removed.

Page 3

REVIEWER:

Footnote 14 makes the text look like 1¹⁴: please make that reference parenthetical. This applies elsewhere in the paper as well where footnotes are connected to numbers.

ANSWER: Parentheses are added wherever needed.

Page 4

REVIEWER:

Define BK1, BK3, RK1, RK3 and give a citation (if this isn't the first time these abbreviations have been used in this source; otherwise, reference Figure 1a).

ANSWER: A note and reference are now added in the caption of new Figure 2a.

REVIEWER:

Are there any reasons besides purely sensitivity why the redshifted component of the outflow would not have detectable polarization? Are there known physical differences between the two lobes of the outflow? It is particularly intriguing that the CARMA observations of the inferred magnetic field show much more polarized emission along the redshifted outflow lobe, where no SiO polarization is detected.

ANSWER: Good question! I have no other clear answer for that. Notice that, however, jet and outflow are different components with different materials (see explanation above)

Page 6

REVIEWER:

Please describe where the velocity dispersion comes from: for the Davis-Chandrasekhar-Fermi method to make sense, this velocity dispersion should ideally be from turbulence, and thus it is important to know which tracer it is. If I am correct in assuming that the quoted ~15 km/s is the FWHM of the SiO velocity distribution in Figure 8 of Lee et al. 2009, then that is not necessarily a good tracer, since the SiO width is dominated by the outflow, not by turbulent motions that would be disturbing an ordered magnetic field.

ANSWER: It is indeed from the FWHM of the SiO velocity distribution in Figure 8a of Lee et al. 2009 towards knot BK1. This velocity width can be produced by a turbulent motion due to localized shocks in the knot. A note is added.

REVIEWER:

Additionally: please state that $2.8e6$ /cc is the volume density of the jet (assuming that my interpretation of that number from Lee et al. 2010 is correct). Otherwise it might be confused with, for example, the volume density of the gas+dust in the dense core.

ANSWER: A note is added for "the volume density of the jet"

REVIEWER:

You mention the very reasonable idea of following up with SiO(5-4). However,

you say on page 3 that SiO(8-7) is the brightest transition in HH 211. How much fainter is SiO(5-4), and what is an approximate time estimate for how long it would take to get a robust ($>3\sigma$) detection of polarization in both the red- and blueshifted lobes in the HH 211 outflow in the fainter J=5-4 transition? Is it doable in what is now considered a "reasonable" amount of ALMA time?

ANSWER: The polarization degree in SiO J=5-4 is not known yet and can be assumed to be $\sim 1\%$, as in the fainter region of the jet in SiO J=8-7.

Since

the SiO emission is close to be optically thick, we can assume that SiO 5-4 has a similar brightness temperature, and thus a factor of ~ 3 fainter in Jy/beam. In SiO J=8-7, the emission peaks are ~ 200 mJy/Beam in a $0.21'' \times 0.14''$ beam, and thus SiO J=5-4 emission will have a peak of ~ 70 mJy/Beam in the same beam. In order to have a peak of 7 sigma detection as we have here for a robust detection, we need a noise level of ~ 0.10 mJy/beam. The corresponding integration time on source would be ~ 3 hrs. Thus, the answer is yes. However, a realistic polarization model is needed to check our estimate. Thus, this is just for your reference and I will not include this in the paper.

***** Methods: Observations *****

REVIEWER:

¶2: the noise levels referred to are in the spectral-line data (not the dust continuum), correct? Best to say that specifically.

ANSWER: Yes, a note is added.

REVIEWER:

¶3: the note about ALMA's restrictions is correct. However, this statement makes me wonder: is all of the detected polarized emission within a radius of $\sim 2''$ from the source position? Or are you masking robustly detected polarization further out in the jet? If polarization IS detected far out in the jet (i.e., where you can compare with the 2014 SMA results), I believe it should still be shown, just with stated caveats regarding deviations in polarization position angle and polarization fraction further out in the ALMA primary beam.

ANSWER: No SiO polarization detected beyond $2''$ of the central source. A note is added to make this clear.

REVIEWER:

Please describe how you performed self-calibration on the image, if you did in fact do so (and if you didn't, then you definitely should to increase image fidelity).

ANSWER: Continuum intensity map of the central source was used to

self-calibrate the data. A note is added.

REVIEWER:

Please state whether the images in Figures 1 and 2 are primary-beam corrected.

ANSWER: No primary beam correction is applied because we only focus on the inner part of the jet where the polarized emission can be mapped properly. In addition, no polarization is detected beyond 2" of the central source.

A
note is added.

***** Figures *****

FIGURE 1

REVIEWER:

-Mention in the figure caption what A, B, and the BKs/RKs are

ANSWER: Mentioned as suggested.

REVIEWER:

-I recommend using a perceptually uniform sequential colorscale for both the

- polarized intensity colorscale and the line-segment colors

-Example: https://matplotlib.org/examples/color/colormaps_reference.html

ANSWER: Thanks for your suggestion. Modified "plasma" color scale has been adopted.

REVIEWER:

-Please do not use light green or cyan on a white background: they are very
- hard to see

ANSWER: Modified version of "plasma" color scale is now used for the line segments as well

FIGURE 2

REVIEWER:

-Caption error: "Red image with the red contours shows the REDSHIFTED jet
- component."

ANSWER: Thanks. Corrected

REVIEWER:

-Please make sure that the line segments are plotted at the Nyquist

- frequency (max 2 per beam) along both the major and minor axes of the

- synthesized beam. It appears that the line segments are a bit too close

- together at the moment (though I may be wrong).

ANSWER: Updated with Nyquist sampling

=====
=

Reviewer #2 (Remarks to the Author):

This paper reports the detection and mapping of SiO line polarization towards the jet of a low-mass protostar. This result gives valuable information on the distribution of the magnetic field towards this jet, since this linear polarization is expected to be aligned either parallel or perpendicular to the projection of the magnetic field on the plane of the sky. This is one of the best cases of protostellar jets in which polarization has been mapped, and likely the best example in a low-mass protostar. It is also the case in which the polarization/magnetic field has been mapped in the jet at closest distances to the star (a few hundred au). These results would make the paper suitable for publication in Nature Communications. However, in its present form, the manuscript presents severe problems that should be fully addressed before consideration for publication.

Main issues:

REVIEWER:

1. The manuscript apparently reports observations of the HH 211 jet, located in the Perseus molecular cloud. The coordinates of the center of the HH 211 jet are approximately RA(J2000) = 03 43 57, DEC(J2000) = +32 00 50. However, all the images in the paper are given in offsets relative to the central source of the jet, and the absolute coordinates of this central source, which are given only in the caption of Fig 1, RA(J2000) = 05 43 51.4086, DEC(J2000) = -01 02 53.147, correspond to a completely different region of the sky. I concluded that the coordinates given in the paper actually correspond to the HH 212 jet. Thus, which one of the two objects, HH 211 or HH 212, has been observed? This is a serious error that compromises the credibility of the observational data presented in the paper.

Fortunately, the orientation of the two jets is very different. The HH 211 jet is oriented in a SE-NW direction while HH 212 is oriented in a SW-NE direction, suggesting that the images presented in the paper actually correspond to the right object. However, some authors prefer to rotate the HH 212 jet by 90 deg, in which case the two jets look very similar. The images presented in the paper do not indicate the direction of the Position Offsets, making more difficult the correct identification. Are the Offsets in the RA, DEC directions? Where is the North?

The authors must completely dissipate any doubt on the identification of the source of their observations before further consideration for publication of this paper. I suggest to use absolute coordinates instead of offsets in at least one of the images presented. Further details on the observations and observational setup should be given in the Methods section (e.g, give the project number, position of the phase center, etc). The physical properties of the central protostar and its close environment are obtained from the literature, including papers with no absolute coordinates (in a number of cases with several authors in common with those of this paper); because the differences between the central sources are not as obvious as in the jets,

it should be double-checked that the parameters adopted really correspond to the driving source of HH 211 and not to that of HH 212.

ANSWER: Really sorry for this embarrassing mistake. It is indeed HH 211. I have corrected the coordinates and adopted the (RA, Dec) coordinates in new Figure 1 and Figure 2a (was Figure 1a) to be clear. The project number and the position of the phase center are now added in Observations.

REVIEWER:

2. Related with the issue of the position of the source, the authors may want to know that a distance of 321 ± 10 pc has recently been proposed for IC 348, where the HH 211 jet is located (see Ortiz-Leon et al. 2018, arXiv:1808.03499v1). This estimate has been obtained from new VLBI data and seems to be in better agreement with Gaia DR2. Ortiz-Leon et al. suggest that the previous estimate by Hirota et al. (2011) (the value used by the authors, although not referenced), obtained from water masers, could be affected by variability and problems in the identification of the maser features. This issue must be discussed and adoption of the new distance should be considered.

ANSWER: Thanks for the info. The new distance is now adopted.

REVIEWER

3. The authors introduce their results in a way that may suggest that it is the first mapping of the polarization/magnetic field in a protostellar jet (3rd paragraph), and that there was no previous evidence of synchrotron radiation in a jet from a low-mass protostar (4th paragraph). This is not correct, as there were previous results providing convincing evidence of both things, and this should be properly explained.

Carrasco-Gonzalez et al. (2010) discuss in detail the strength, direction and spatial distribution of the magnetic field, and give quite convincing evidence that it is intrinsically associated with the HH80-81 jet. For example, Figure 3 in that paper shows that the magnetic field inferred from the polarization of the radio jet is very well aligned along the jet axis, while the direction of the magnetic field in the ambient cloud inferred from the dust emission presents a large dispersion. This information was derived from linearly polarized synchrotron emission in a way similar to the well-studied AGN jets. For non-relativistic jets, as is the case of the HH80-81 jet, such an approach has the advantage that the magnetic field projected on the plane of the sky is perpendicular to the polarization, with no ambiguity. Carrasco-Gonzalez et al. favor an helical geometry for the magnetic field, again similar to what is expected in AGN jets. Given the large size of the HH 80-81 jet, the angular resolution in that paper (corresponding to about 1500 au) is sufficient to map the distribution of the magnetic field with respect to the jet, but at distances $> 30,000$ au from the exciting source, as noted by the authors of the present manuscript. Therefore, the main novelty of the results presented here is that polarization is mapped for the first time towards the jet of a low mass protostar, and at much closer distances from the star than in previous

studies.

ANSWER: Thanks for the useful explanations. In Carrasco-Gonzalez et al. (2010), since dust polarizations were only detected at the base near the central source, there was no information about the field morphology in the ambient cloud at large distance around the jet. Interestingly, however, the dust polarizations detected around the central source appears like the center of a pinched hour-glass field morphology often seen in other sources. If this is the case, then the field line in the ambient cloud around the jet could also be aligned with the jet axis.

We have added a sentence stating that they favor helical geometry for the magnetic field, similar to what is expected in AGN jets.

The angular resolution in that paper was $13'' \times 8''$ (see their Fig 1 caption) or 22000×14000 au correspondingly at a distance of 1.7 kpc, but not 1500 au as mentioned in your comment. Thus the intrinsic jet, with a radius < 2000 au, was not resolved in their study. Nonetheless, we have removed the sentence about the jet radius.

We have emphasized the novelty of our work as you suggested, i.e., first detection of line polarization towards the jet of a low-mass protostar, and at much closer distances from the protostar than in previous studies. Thanks.

REVIEWER:

The paper by Rodriguez-Kamenetzky et al. (2017) does not seem to contradict the previous paper by Carrasco-Gonzalez et al., as apparently is suggested in the manuscript (actually, both papers have many authors in common). The paper by Rodriguez-Kamenetzky et al. mainly deals with the origin of the relativistic electrons that produce the synchrotron emission rather than with the distribution of the magnetic field. Polarization is not detected in the second paper despite the increased sensitivity. The authors attribute this non-detection to the lower flux density per beam because of the higher angular resolution. Also, the large depolarization effects due to Faraday Rotation in the large bandwidth of the new observations likely have an important contribution. Therefore, the magnetic field distribution cannot be discussed in detail from the results of this second paper. Nevertheless, the higher angular resolution of Rodriguez-Kamenetzky et al. led these authors to suggest that the synchrotron emission in the jet arises from an extended component and from the termination points of the jet. In my opinion this does not exclude that the magnetic field mapped by Carrasco-Gonzalez et al. is intrinsically associated with the jet itself; otherwise, a re-orientation along the jet axis of the ambient cloud magnetic field would be required. Sensitive high-angular resolution observations with a narrow bandwidth would set this issue.

I suggest the 3rd paragraph to be reworded to better reflect the available previous observations, in order to clarify the relevance of the new data.

ANSWER: Thanks for your useful explanation. We have reworded the related sentences in that paragraph accordingly. BTW, as mentioned above, the magnetic field line in the ambient cloud around the jet at that large distance could also be aligned with the jet axis and thus re-orientation may not be needed.

REVIEWER:

In the 4th paragraph, it is not correct to say that "the jets from low-mass protostars do not show any signs of synchrotron radiation". Carrasco-Gonzalez et al. (2010) mention several examples of non-thermal emission in jets from low-mass objects suggestive of synchrotron. Also, the paper by Ainsworth et al. (2014), ApJ, 792, L18 presents tentative evidence for synchrotron emission associated with the jet of the young low-mass star DG Tau, and should be cited.

ANSWER: Thanks for pointing that out. Tentative detections of synchrotron emission for radio jets in L778 VLA5 and DG Tau are now added.

Minor Comments:

REVIEWER:

Paragraph 9: Only poloidal and toroidal magnetic fields are considered. What about an helical magnetic field? Note that what is obtained from the observed direction of the polarization is the projection of the magnetic field on the plane of the sky.

ANSWER: Thanks. A discussion of helical magnetic field is now added.

REVIEWER:

Paragraph 12: A "jet radius" of < 20 au is mentioned. It could be better to use "cylindrical radius" to avoid ambiguities with the launching radius or with the distance from the origin of the jet.

ANSWER: "cylindrical radius" is now used to avoid ambiguities.

REVIEWER:

Methods: It is stated that the maximum recoverable scale is about 1.1 arcsec. This scale is larger than the width of the HH 211 jet but significantly smaller than the extension of both the blue and the red lobes along the direction of the jet axis ($> 3-5$ arcsec). How this observational constraint affects the results presented here? Is it feasible that the polarized emission of a possible extended (> 3 arcsec) structure elongated along the jet axis is partially resolved out by the interferometer? Could this effect "erase" part of the emission and affect the observed direction of the polarization vector? Could this effect be partially responsible for the observed lack of polarization in the redshifted lobe?

ANSWER: Since the jet is knotty with the subknots having a size of $\sim 0.2''$, the detection of polarized emission towards the subknots will not be affected. As you pointed out, since the jet seems to have a smooth

structure greater than the maximum recoverable scale, the polarized emission there, if exists, could be partially resolved out. Since the knots are brighter than the smooth structure, the lack of polarization in the redshifted lobe should be mainly because of not enough brightness temperature. Notes are added about these.

REVIEWER:

Caption of Fig 1: the jet is rotated...  the image of the jet is rotated..

ANSWER: Revised.

REVIEWERS' COMMENTS:

Reviewer #1 (Remarks to the Author):

I have reviewed the revisions by Lee et al. and find the corrections to be sufficient. I recommend this manuscript for publication in Nature Communications once a few more small comments (below) are addressed. I do not need to see a revised version.

I would particularly like to thank Chin-Fei for the useful discussion regarding my 2014 CARMA paper. I am truly embarrassed that I accidentally neglected to remove the CO contamination from some of the dust maps in my CARMA survey, but am thankful for Chin-Fei for finally bringing it to my attention -- this sort of insightful, critical analysis of past work is essential for pushing the field forward.

Many thanks,
Charles L. H. (Chat) Hull

COMMENTS

+ p.4: 2568 au is too precise; make the significant figures consistent with the uncertainty in the distance.

***** FIGURES *****

My only other comments have to do with the updated figures (which look much better, by the way!). My mild color-blindness is the cause of the problems, as usual.

+ Fig 1: please choose two more contrasting colors to make the jet stand out from the outflow. Perhaps blue or red for the CO and yellow for the jet? The black background is excellent.

+ Figs 2(c), 3(a), and 5: I'm having trouble seeing some of the vectors.

In 2(c) and 3(a) it's because the highest-SNR vectors are yellow, and are on a white background. This could be solved by adding a thin black outline to all vectors.

In Fig 5 it's the opposite problem: the red/purple-ish vectors are hard to see on top of the blueshifted outflow (the only vectors I can see well are the yellowish ones). Since this is just the "possible fields" plot, I don't think it's essential to maintain the SNR-based colorscale for the vectors (kind of like Figure 4, where the vectors are all the same color, since the point of that figure is just to differentiate between the dust and SiO polarization). Perhaps just switch to vectors that are yellow (or some other bright color) in this figure.

Reviewer #2 (Remarks to the Author):

The paper has been very much improved in this revised version. It is essentially ready for publication. I only have a few very minor comments:

Page 2, 2nd par: Regarding previous linear polarization results on HH80-81, it is said that the polarization orientations are roughly aligned with the jet axis. Observed polarization vectors in the HH80-81 jet are perpendicular to the direction of the jet. What is aligned with the jet axis is the component in the plane of the sky of the magnetic field.

I want to add here that in my previous report there was an unfortunate typographical error as the angular resolution of the HH80-81 observations should read 15000 au instead of 1500 au (I said 15000 au instead of 20000 au because recent estimates suggest a closer distance of 1.4 kpc instead of 1.7 kpc, but this difference is irrelevant here). What I meant to say was that in those observations the jet was resolved longitudinally, providing the distribution of polarization and magnetic field along the jet, not transversally. Sorry for the confusion. As in HH 211, the HH80-81 jet is only marginally resolved in the transversal direction (see Fig 3 of Carrasco-Gonzalez et al). Note also that it is not necessary to resolve transversally the jet to be sure that the observed linear polarization is intrinsic to it because this emission originates from ionized material, which should not be present in the ambient gas. This is a situation similar to your HH 211 SiO observations, making both HH80-81 and HH 211 two clear examples. CO is more ambiguous because this molecular line can originate either from the jet or from ambient gas.

Page 4, 1st line: Since jets are indeed (very collimated) outflows, to avoid ambiguity I suggest to say: "is intrinsically different from the CO outflow".

2nd par: If the field of view is given as ~ 8 arcsec, then ~ 2600 au seems more adequate than 2568 au for the corresponding physical scale.

"The jet is marginally resolved"  "The jet is marginally resolved in the transversal direction" (or, "The jet width is marginally resolved") to avoid confusion, as usually a "marginally resolved source" means that the source appears as nearly unresolved in any direction.

"The radius of the jet"  "The cylindrical radius of the jet"

Page 5, 1st par, last line: "in smaller velocity channels"  "in narrower velocity channels" (or "in channels with a smaller velocity width")

Page 8, 1st line: the observed orientations of the polarization ARE the projections of the magnetic field...  the observed orientations of the polarization GIVE the projection of the magnetic field...

2nd par: Could you please clarify the meaning of "updated for the new jet radius"? Do you refer to an update because of the new distance, because now you obtain a jet width different from that measured by Lee et al. (2009), or because of any other reason...

Page 10, 1st par: subknots, which has a size  subknots, which have sizes (?)

Figs 2 and 3: I agree that using a perceptually uniform sequential colorscale is better. However, the selected color palette makes almost impossible to see the yellow line segments on a white background. This is particularly unfortunate because the yellow line segments are those with a higher S/N. Also, it seems that some of these yellow segments partially hide the darkest ones and the information on their length is lost. Since background is white, perhaps the colorscale should be reversed, with higher intensities and higher S/N corresponding to darkest colors (or use a black background as in Fig 5. Alternatively, use only the darkest part of the color scale (e.g, from black to up to orange), avoiding yellow...

Guillem Anglada

Dear Chat and Guillem,

Thank you very much for your further constructive comments. We have revised the paper accordingly, with the changes in bold-faces. Really sorry for the difficult viewing of the line segments in Figs 2 and 3. We have added a gray background to better show the line segments. We have also changed the contours to blue for blueshifted jet component and red for redshifted jet component.

Best Regards,
Chin-Fei

=====
REVIEWERS' COMMENTS:

Reviewer #1 (Remarks to the Author):

I have reviewed the revisions by by Lee et al. and find the corrections to be sufficient. I recommend this manuscript for publication in Nature Communications once a few more small comments (below) are addressed. I do not need to see a revised version.

I would particularly like to thank Chin-Fei for the useful discussion regarding my 2014 CARMA paper. I am truly embarrassed that I accidentally neglected to remove the CO contamination from some of the dust maps in my CARMA survey, but am thankful for Chin-Fei for finally bringing it to my attention -- this sort of insightful, critical analysis of past work is essential for pushing the field forward.

Many thanks,
Charles L. H. (Chat) Hull

COMMENTS

REVIEWER:

+ p.4: 2568 au is too precise; make the significant figures consistent with the uncertainty in the distance.

ANSWER: Revised to 2600 au.

***** FIGURES *****

My only other comments have to do with the updated figures (which look much better, by the way!). My mild color-blindness is the cause of the problems, as usual.

REVIEWER:

+ Fig 1: please choose two more contrasting colors to make the jet stand out from the outflow. Perhaps blue or red for the CO and yellow for the jet? The black background is excellent.

ASNWER: Tried with blue/red and yellow as suggested, but it didn't make the jet stand out better from the outflow. Thus, I chose gray and red, which should make the jet stand out better, as shown in Figure 1.

REVIEWER:

+ Figs 2(c), 3(a), and 5: I'm having trouble seeing some of the vectors. In 2(c) and 3(a) it's because the highest-SNR vectors are yellow, and are on a white background. This could be solved by adding a thin black outline to all vectors.

ANSWER: In Fig 2c and Fig 3, gray background is now added to show the line segments better.

REVIEWER:

In Fig 5 it's the opposite problem: the red/purple-ish vectors are hard to see on top of the blueshifted outflow (the only vectors I can see well are the yellowish ones). Since this is just the "possible fields" plot, I don't think it's essential to maintain the SNR-based colorscale for the vectors (kind of like Figure 4, where the vectors are all the same color, since the point of that figure is just to differentiate between the dust and SiO polarization). Perhaps just switch to vectors that are yellow (or some other bright color) in this figure.

ANSWER: Thanks, yellow line segments are now used as suggested. The scale bar is removed to conform with the journal policy.

=====
Reviewer #2 (Remarks to the Author):

The paper has been very much improved in this revised version. It is essentially ready for publication. I only have a few very minor comments:

REVIEWER:

Page 2, 2nd par: Regarding previous linear polarization results on HH80-81, it is said that the polarization orientations are roughly aligned with the jet axis. Observed polarization vectors in the HH80-81 jet are perpendicular to the direction of the jet. What is aligned with the jet axis is the component in the plane of the sky of the magnetic field.

ANSWER: Thanks you for pointing out the mistake. We have revised "aligned with" to "perpendicular to".

REVIEWER:

I want to add here that in my previous report there was an unfortunate typographical error as the angular resolution of the HH80-81 observations should read 15000 au instead of 1500 au (I said 15000 au instead of 20000 au because recent estimates suggest a closer distance of 1.4 kpc instead of 1.7 kpc, but this difference is irrelevant here). What I meant to say was that in those observations the jet was resolved longitudinally, providing the distribution of polarization and magnetic field along the jet, not transversally. Sorry for the confusion. As in HH 211, the HH80-81 jet is

only marginally resolved in the transversal direction (see Fig 3 of Carrasco-Gonzalez et al). Note also that it is not necessary to resolve transversally the jet to be sure that the observed linear polarization is intrinsic to it because this emission originates from ionized material, which should not be present in the ambient gas. This is a situation similar to your HH 211 SiO observations, making both HH80-81 and HH 211 two clear examples. CO is more ambiguous because this molecular line can originate either from the jet or from ambient gas.

ANSWER: Thanks for your explanations and I agree with you.

REVIEWER:

Page 4, 1st line: Since jets are indeed (very collimated) outflows, to avoid ambiguity I suggest to say: "is intrinsically different from the CO outflow".

ANSWER: Revised accordingly.

REVIEWER:

2nd par: If the field of view is given as ~ 8 arcsec, then ~ 2600 au seems more adequate than 2568 au for the corresponding physical scale.

ANSWER: Revised to 2600 au. Thanks.

REVIEWER:

"The jet is marginally resolved"  "The jet is marginally resolved in the transversal direction" (or, "The jet width is marginally resolved") to avoid confusion, as usually a "marginally resolved source" means that the source appears as nearly unresolved in any direction.

ANSWER: Thanks. Revised accordingly.

REVIEWER:

"The radius of the jet"  "The cylindrical radius of the jet"

ANSWER: Revised accordingly

REVIEWER:

Page 5, 1st par, last line: "in smaller velocity channels"  "in narrower velocity channels" (or "in channels with a smaller velocity width")

ANSWER: Revised to "in narrower velocity channels"

REVIEWER:

Page 8, 1st line: the observed orientations of the polarization ARE the projections of the magnetic field...  the observed orientations of the polarization GIVE the projection of the magnetic field...

ANSWER: Revised accordingly.

REVIWER:

2nd par: Could you please clarify the meaning of "updated for the new jet

radius"? Do you refer to an update because of the new distance, because now you obtain a jet width different from that measured by Lee et al. (2009),
o
because of any other reason...

ANSWER: The measured jet width in arcsec is roughly the same as before. Thus, the difference in jet radius in au is mainly because of the new distance. A note is added.

REVIEWER:

Page 10, 1st par: subknots, which has a size  subknots, which have sizes (?)

ANSWER: Revised to "have a size"

REVIEWER:

Figs 2 and 3: I agree that using a perceptually uniform sequential colorscale is better. However, the selected color palette makes almost impossible to see the yellow line segments on a white background. This is particularly unfortunate because the yellow line segments are those with a higher S/N. Also, it seems that some of these yellow segments partially hide the darkest ones and the information on their length is lost. Since background is white, perhaps the colorscale should be reversed, with higher intensities and higher S/N corresponding to darkest colors (or use a black background as in Fig 5. Alternatively, use only the darkest part of the color scale (e.g, from black to up to orange), avoiding yellow...

ANSWER: In Fig 2c and Fig 3, gray background is now added to show the line segments better.